# The Framework Tax:
# Disparities Between Inference Efficiency in NLP Research and Deployment

**Jared Fernandez**[1]    **Jacob Kahn**[3]    **Clara Na**[1]    **Yonatan Bisk**[1]    **Emma Strubell**[1,2]

[1]Language Technologies Institute, Carnegie Mellon University
[2] Allen Institute for Artificial Intelligence    [3]FAIR

{jaredfern, strubell}@cmu.edu, {csna, ybisk}@cs.cmu.edu, jacobkahn@fb.com

## Abstract

Increased focus on the computational efficiency of NLP systems has motivated the design of efficient model architectures and improvements to underlying hardware accelerators. However, the resulting increases in computational throughput and reductions in floating point operations have not directly translated to improvements in wall-clock inference latency. We demonstrate that these discrepancies can be largely attributed to bottlenecks introduced by deep learning frameworks. We denote this phenomenon as the *framework tax*, and observe that the disparity is growing as hardware speed increases over time. In this work, we examine this phenomenon through a series of case studies analyzing the effects of model design decisions, framework paradigms, and hardware platforms on total model latency. Code is available at https://github.com/JaredFern/Framework-Tax.

## 1   Introduction

Natural language processing systems have benefited from improvements in performance driven by scaling of training data and number of model parameters (Kaplan et al., 2020; Alabdulmohsin et al., 2022; Tay et al., 2021, 2022). However, the accompanying increases in computation raise concerns as to the efficiency of these systems due to associated environmental costs of development and deployment (Schwartz et al., 2020; Strubell et al., 2019).

In particular, efficiency is especially important in inference settings where models are used repeatedly and at scale, in contrast to training which poses a single upfront computational cost. For example, Meta reports that inference workloads make up 70% of their AI power consumption, with the remaining 30% due to training and development (Wu et al., 2022), while Google attributes 60% of their ML energy consumption to inference (Patterson et al., 2022). Inference is also estimated

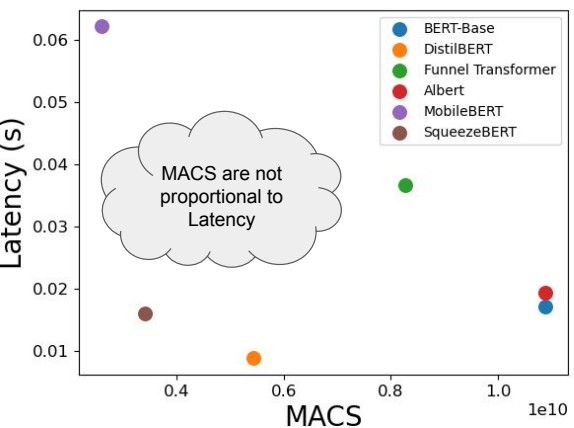

Figure 1: Latency as a function of a model's multiply accumulate operations (MACs) on an Nvidia 2080ti GPU. Expected relationships do not hold; model complexity and hardware capabilities fail to predict latency due to framework-boundedness.

to make up 80 to 90% of ML cloud computing demand (Barr, 2019; Leopold, 2019). In these settings, metrics of model speed such as latency and throughput are essential for inference workloads that are subject to real wall-clock time constraints such as real-time natural language generation and automated speech recognition (Reddi et al., 2020).

These concerns have motivated research in designing more efficient neural network model architectures and faster hardware accelerators. In the past five years alone, the number of papers that mention the terms *efficient* or *efficiency* in top machine learning venues has grown by over 2.5x and even more so at venues in natural language processing, increasing by 8.3x in the same span.[1] This has spurred innovations in the design of efficient neural network architectures for language aiming to reduce the number of trainable model parameters, floating point or multiply-accumulate operations (MACs) (Iandola et al., 2020; Dai et al., 2020; Sun

---

[1]Based on publications at ML (ICLR, ICML, NeurIPS) and NLP conferences (ACL, EMNLP) between 2017 and 2022

et al., 2020). Over the same period, GPU hardware accelerators have seen similar performance gains with the number of floating point operations per second (FLOPS) growing by over 175%.[2]

Despite this progress, *the supposed gains offered by higher performance hardware and more efficient models are often not realized in inference settings, where wall-clock model speed has not reliably improved*, as seen in Figure 1. We show that this misalignment is primarily attributable to overhead incurred by deep learning frameworks used to implement and execute models. In the past, the execution of large neural networks common to natural language processing has been assumed to be compute-bounded by computationally intensive tensor operations (Li et al., 2020). However, as the speed of hardware increases, the overhead introduced by the deep learning frameworks used to implement and deploy these models is no longer negligible and imposes bottlenecks on inference.

In this work, we conduct a systematic investigation in which we show that improvements to neural network architectures and hardware have not translated into reductions in inference latency. We show that this breakdown is largely due to overhead introduced by deep learning frameworks. We refer to this phenomenon as the *framework tax*, and show that it exists across all deep learning framework paradigms (e.g. eager execution, just-in-time, and ahead-of-time compilation).

At small batch sizes and sequence lengths, we show that fixed framework overhead dominates inference time leading proxy measurements of model efficiency, such as MACs and parameter count, to breakdown as predictors of inference latency. Furthermore, we note that existing efforts to improve model efficiency by designing lower FLOP model architectures and faster GPU kernels do not reduce latency, when total execution is bound by fixed-cost framework overhead. Moreover, we note that as hardware performance increases, NLP systems will become *increasingly* framework-bound at larger batch sizes.

An exhaustive comparison of the rapidly growing space of models and hardware platforms is out of scope, so we instead identify the most popular model components, inference environments, and hardware accelerators for testing (i.e. isolating the cases most likely to mislead a practitioner). We analyze the performance of transformer and convolutional neural network models in eager execution PyTorch, just-in-time compiled TorchScript, and ahead-of-time compiled ONNX runtime using a CUDA execution provider. We perform our study across seven different GPUs from the Pascal, Turing, and Ampere Nvidia GPU microarchitectures.

Based on our findings, we provide a series of recommendations for NLP researchers and practitioners presented through a collection of case studies. Among these, we recommend usage of static or ahead-of-time inference runtimes when batch sizes are small as they can substantially reduce framework overhead. Alternatively, when using eager execution-based frameworks for inference, we recommend increasing model width or batch size at no cost to latency and take advantage of performance gains associated with increased model capacity (Zagoruyko and Komodakis, 2016). For example, hidden dimensions in self-attention and fully connected layers can be doubled to increase model capacity without affecting latency, implying that model designers have an extra degree of freedom often overlooked when designing around parameters or FLOPs. We hope that our analysis and recommendations will help bridge the gap between efficient NLP research and practice.

## 2 Related Work

### 2.1 Efficiency Metrics & Cost Indicators

Previous efforts to report efficiency often utilize proxy metrics for amount of computation, such as the number of floating point (FLOPs) or multiply-accumulate (MACs) operations (Schwartz et al., 2020). Similarly, number of trainable parameters is a frequently reported as a proxy for memory utilization (Lan et al., 2019). Unfortunately, these proxy metrics are often not predictive of realworld efficiency. For example, total FLOPs does not account for the varying extent to which different operations can be parallelized and techniques such as weight tying can reduce parameter counts without reducing the amount of required computation (Lan et al., 2019). From the perspective of device utilization, hardware and model FLOPs utilization (Chowdhery et al., 2022) are reported as the ratio between observed FLOPs per second and a hardware platforms peak theoretical FLOPs.

Previous works examining the relationship between efficiency measures showed that different cost indicators do not correlate well with each other during neural network training (Dehghani et al.,

---

[2]Per theoretical FLOPS of Nvidia GPUs (2017-2022).

2021). In particular, it has been hypothesized that discrepancies between FLOPs and wallclock inference latency is primarily are primarily compute bounded by kernel execution or memory-bound by data movement as opposed to framework bottlenecks (Langerman et al., 2020). These previous works have largely focused on convolutional neural networks (CNNs) in computer vision. We extend these analyses to the inference setting and study transformer-based neural networks for NLP, and show that FLOP-based proxy metrics breakdown for due to additional performance bottlenecks introduced by deep learning frameworks.

## 2.2 Efficient Model Design

Desire to develop computationally efficient models for language processing has led to the development of a variety of model architectures that achieve comparable task performance under fixed FLOP budgets. For example, compression of input sequences and intermediate representations has been used to reduce the computational cost of long text sequences (Dai et al., 2020; Goyal et al., 2020) and distillation has been used to reduce model size (Sanh et al., 2019; Hou et al., 2020). Other work has sought to design efficient model architectures by developing low-FLOP substitutes for standard, dense self-attention and convolution operations (Iandola et al., 2016; Zhang et al., 2018; Sandler et al., 2018; Sun et al., 2020; Xiong et al., 2021; Wang et al., 2020b).

Additionally, direct efficiency metrics, such as wall-clock latency and energy usage have been incorporated into the objectives of neural architecture search (NAS) and AutoML methods (Wu et al., 2019; Tan et al., 2019; Wang et al., 2020a). Manual inspection of the learned models shows that NAS often implicitly learns to take advantage of supported parallelism, learning wider architectures on GPU devices and deeper models on CPU devices (Cai et al., 2018; Tan et al., 2019; Sandler et al., 2018). However, it is often impossible to control for the hardware systems used for collecting these metrics leading to conclusions that may not generalize across deployment settings.

## 2.3 Platform Performance Analysis

Efforts to establish common benchmarks leverage reference models and hardware platforms with target latency or accuracy (Reddi et al., 2020; Zhou et al., 2020). Although these efforts have led to improvement in end-to-end latency, they often abstract away the underlying frameworks, compilers, backends, and hardware platforms. While general improvements in hardware and software kernels may lead to improvements across all models, it has been argued that solely focusing on performance optimization of a limited set of model architectures and runtimes may lead to overspecialization (Hooker, 2021).

Previous analysis of the computational properties of hardware accelerators has largely focused on the training setting in which larger kernels and batch sizes hide framework overhead that emerges in the inference setting (Wang et al., 2020c; Zhu et al., 2020, 2018). Other analyses of end-to-end systems analyses has primarily focused on domain-specific applications in reinforcement learning and recommendation systems (Gleeson et al., 2021; Lin et al., 2022), where simulation and memory access dominate execution time. Additionally, these prior efforts are restricted to small sets of reference models and have not directly examined the relationship between model architectures and platforms.

## 3 Preliminaries

### 3.1 Neural Network Frameworks

To take advantage of massively parallel hardware accelerators, inference with variable length text and speech sequences are padded to fixed length tensors that are processed with neural network frameworks. These frameworks provide implementations and APIs for tensor operations, gradient calculation, and construction of neural network computational graphs. Frameworks generally fall into the following design paradigms (Kahn et al., 2022):

**Eager Execution:** The computational graph is constructed from a series of operations that are executed as soon as called from an interpreter. Examples include: PyTorch (Paszke et al., 2019) and Chainer (Tokui et al., 2015).

**Deferred Execution:** A series of operations are defined and executed on sample data to generate a dataflow graph that can then be just-in-time (JiT) compiled. Examples include: TorchScript, Jax (Bradbury et al., 2018), Theano (Al-Rfou et al., 2016), Caffe (Jia et al., 2014).

**Static:** The computational graph is pre-defined, compiled, and executed inside a specialized runtime; allowing for aggressive, global ahead-of-time (AoT) compiler optimizations. Examples include:

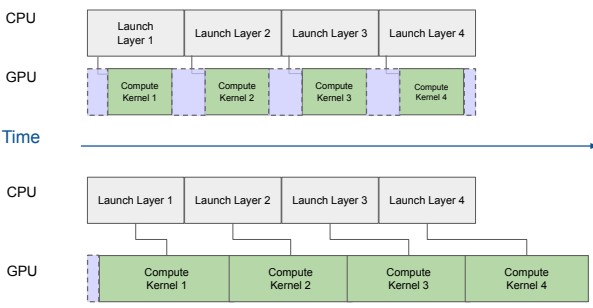

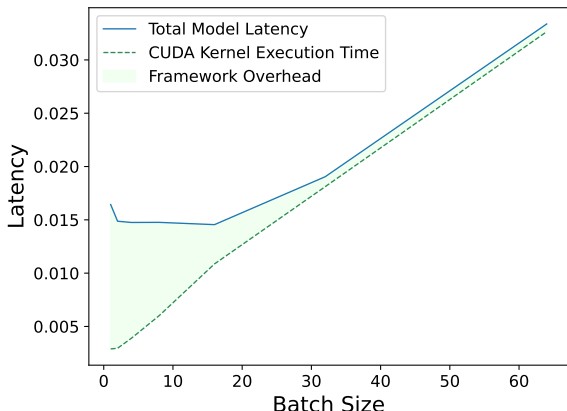

Figure 2: Profiles where execution is *framework bound* by CPU kernel dispatch operations (above) and *compute-bound* by GPU kernel operations (below). Small compute kernels occur in inference at lower batch sizes. Boxes with dashed lines represent *framework overhead*.

Figure 3: Framework overhead of BERT-Base in PyTorch for various batch sizes on an RTX-8000. Although small batch sizes require fewer FLOPs, these reductions do not translate to speedups in latency. Framework overhead is substantial at low batch sizes but only a small constant at large batch sizes.

ONNX Runtime, TensorFlow 1.0, MXNet, TensorRT, and TVM (Chen et al., 2015, 2018).

Eager execution frameworks compute each layer as separate operations which each incur additional overhead from CPU kernel dispatches. To accelerate execution, deferred execution and static inference frameworks compile the neural network's computational graph to combine multiple operations together (e.g. fusing attention layers and GELU activations) or remove unnecessary computation (i.e. scalar folding). Removal of kernel launch operations can reduce the memory footprint and result in more efficient kernels, thus decreasing overall framework overhead.

While deferred and static frameworks are commonly used in deployment settings, the NLP research community relies heavily on eager mode frameworks during the development of new models for their ease of use. This further exacerbates the community divide, where models are designed under different assumptions than deployed.

### 3.2 Framework Overhead

Deep learning frameworks asynchronously dispatch computation for execution on highly parallelized hardware accelerators, as shown in Figure 2. For sufficiently large compute kernels, such as those during training, models achieve near maximum GPU utilization – measured as the difference between total execution time and active GPU time (Zhu et al., 2018). However, during inference, smaller input sizes lead to suboptimal GPU utilization as the rapid executing kernels do not saturate the fixed cost framework overhead incurred from CPU operations such as kernel launch, graph construction, control flow, and device synchronization;

See Figure 3 (Aminabadi et al., 2022).

Kernel serialization with optimizations such as CUDA Graphs remove the overhead from multiple kernel dispatches by capturing and replaying the entire computational graph as a single operation. However, kernel serialization requires that models are graph safe (i.e. static shapes and static control flow). This can pose challenges for NLP systems which often leverage dynamic computational graphs to deal with variable length sequences, parse tree depths, and batch sizes (Looks et al., 2017).

When the execution of GPU kernel computation is largely blocked by CPU framework operations such as kernel dispatches, the model's execution becomes *framework-bound*. In this setting, latency is constant regardless of batch size or number of MACs computed. For settings where latency is dependent on the execution of computational kernels and data movement, models are *compute-bound*.

## 4 Experiments

We evaluate models frameworks from each of the major paradigms: eager execution PyTorch, deferred execution TorchScript, and statically compiled ONNX Runtime with a CUDA backend.

We use PyTorch 1.12.1 with CUDA 11.6 and Python 3.8.13. We use ONNX Runtime 1.7.0 with CUDA 11.1.1 and cuDNN v8.0.4.3. Baseline experiments are run on a compute node with an Nvidia RTX-8000 GPU and an Intel Xeon E5-2630 CPU with 32 GB of DDRAM memory.

We measure latency, GPU utilization over a

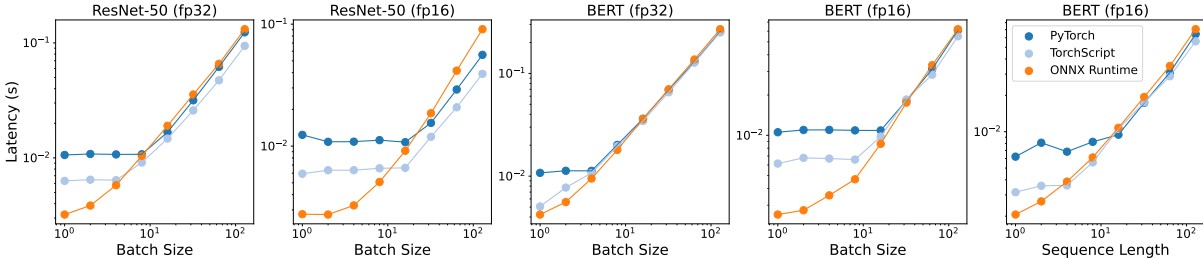

Figure 4: Latency vs. (batch size and sequence lengths) for baseline models in FP16 and FP32 on RTX-8000. Framework boundedness exists for all models at small input sizes where framework overhead dominates runtime and results in constant latency regardless of input size. Framework overhead is most prominent in smaller models executed in half precision on slower frameworks.

range of batch sizes to capture common inference use cases as stated in (Reddi et al., 2020): single example inferences is common in streaming on-device applications, such as autocomplete and automatic speech recognition; large batch inference is typical in offline server settings.

We simulate text data by randomly generating token sequences of length 128, as commonly used in sentence classification tasks (Liu et al., 2019; Izsak et al., 2021). We report averaged metrics from 100 forward passes after 10 warm-up passes to initialize model weights and data. GPU core and memory utilization are measured with the Nvidia Nsight Compute and Nvidia Management Library.

We select BERT-Base (Devlin et al., 2018) and ResNet-50 (He et al., 2016) as representative models for encoder-only and convolutional neural network architectures commonly used for sentence and image classification tasks (Reddi et al., 2020; Janapa Reddi et al., 2022). We evaluate model architectures for natural language generation, speech, and vision and show that they all observe framework bound behavior in Appendices C, D and E.

**Analysis** The number of FLOPs required for model inference scales with the input batch size and sequence length. As such, one would expect that latency scales accordingly as well. However, as seen in Figure 4, models exhibits framework boundedness for both small sequence lengths and batch sizes, where latency is constant regardless of input size.

Computation with mixed and half precision (FP16) often increases training throughput over single precision (FP32), we observe that framework overhead results in latency bottlenecks regardless of the precision during inference. As half precision computation is faster due to reduced data movement, GPU kernel execution time takes longer

to overtake fixed framework overhead. As a result, half precision inference is framework bound for larger batch sizes. For inference with larger compute-bound batch sizes, latency observes expected speedups from using half precision.

Although models based on convolutions (ResNet-50) and self-attention (BERT-Base) operations both exhibit framework-bound behavior, they transition to compute boundedness at different batch sizes. The difference in model behavior can be attributed to differences in the rate at which compute kernels overtake CPU dispatch operations. For the well-optimized operations (Conv2Ds and GEMMs) that make up ResNet-50 and BERT, the time per FLOP is reasonably consistent.

### 4.1 Framework Design Decisions

In Figure 4, we observe that frameworks from all execution paradigms exhibit framework bound behaviors. However, deferred execution TorchScript and static ONNX Runtime, which support computational graph compilation (e.g. operator fusion), exhibit less framework overhead and provide speedups over eager PyTorch. These increases are especially pronounced at low batch sizes where inference is framework-bound. For batch size 1, TorchScript and ONNX provide an average FP16 speed up of 34.16% and 71.38% over PyTorch, respectively. As batch sizes increase and models become compute bound, there is minimal difference in latency across frameworks as the majority of execution time is spent on kernel execution.

Additionally, we consider both static, serialized CUDA graphs and PyTorch BetterTransformer framework optimizations in Figure 5. BetterTransformer provides speedups through additional kernel fusion and sparse tensor operations that take advantage of sparse sequence lengths and padding tokens

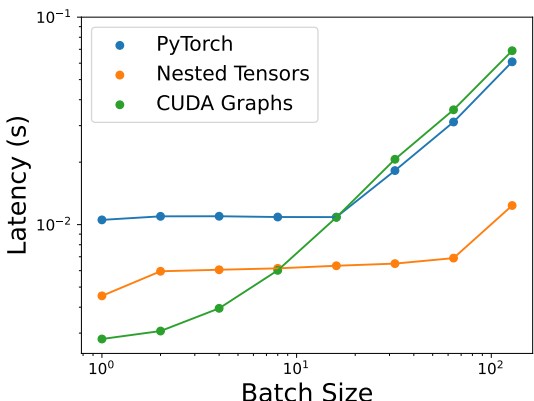

Figure 5: Different framework optimizations lead to latency improvements in different regimes for BERT-Base. CUDA Graph kernel serialization reduces launch overhead in the framework bound regime, whereas sparse computation reduces latency at larger batch sizes.

to remove redundant computation.

To construct sparse inputs, we simulate samples by generating variable length sequences and padding to the maximum sequence length of 128. Sentences are randomly generated according to the sequence length distribution of the Penn Treebank (Taylor et al., 2003), with an average length of 20.92 and a standard deviation of 10.18 tokens.

**Analysis**  Utilization of CUDA Graphs substantially reduce latency at low batch sizes when inference is bounded by framework overhead from kernel launches. However, at larger batch sizes, nested tensor operations can leverage sparsity in padded variable sequence lengths to provide substantial latency reductions when inference is compute-bounded.

## 4.2  Model Design Decisions

We examine a variety of common model architecture design decisions and investigate their alignment with commonly reported efficiency proxies and empirically observed latency.

### 4.2.1  Scaling Model Depth & Width

**Assumption**  Scaling the dimensionality and number of hidden layers is commonly used in NLP and computer vision as a means to explore tradeoffs between model performance and computational requirements (He et al., 2016; Touvron et al., 2021; Zhou et al., 2021). Recent work has shown that model end-task performance scales differently along each axis (Tay et al., 2021, 2022; Nguyen et al., 2021).

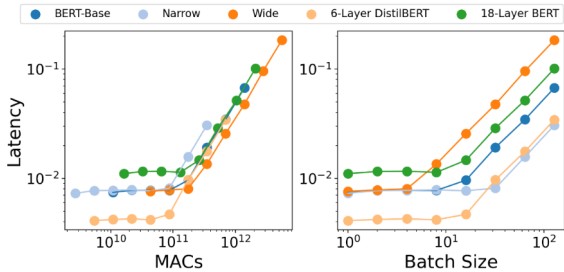

Figure 6: Comparison of latency for BERT variants that scale model *width* and *depth*. Increases in model depth add more framework overhead, whereas increases in model width lead to faster transitions to compute boundedness.

We compare our baseline models to variants that scale both model width and depth. We examine 12-layer BERT-Base against its 6-layer DistilBERT (Sanh et al., 2019) variant and experiment across parameterized BERT models, varying the number of encoder layers as well as width of their fully connected and self-attention layers.

**Analysis**  When scaling model depth, we observe that latency increases in both the framework- and compute-bound regimes as each added layer operation requires an additional CPU-GPU dispatch. Deeper model variants have a larger fixed latency in the framework-bound regime as seen in Figure 6.

Counter-intuitively, wider model variations see *no increase in latency* at low batch sizes. As model execution is framework bound, total runtime is constant despite wider operations requiring more floating point operations. Instead, increased per-layer kernel execution time causes these wider models to become compute-bound at lower batch sizes. In the compute-bound regime, latency scales more rapidly with batch size for wide models.

### 4.2.2  Downsampling and Hierarchical Pooling

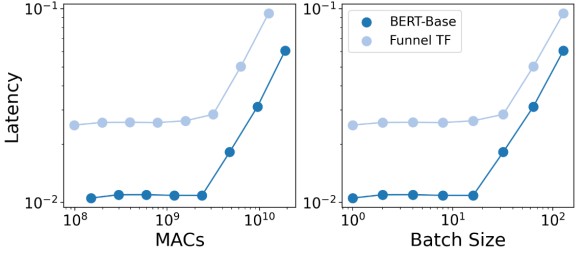

Figure 7: Comparison of BERT and Funnel Transformer latency. Despite using fewer total MAC operations, Funnel is slower than BERT for inference due to the introduction of additional intermediate pooling layers.

**Assumption** Self-attention layers in Transformer architectures are notoriously computationally expensive as their complexity scales quadratically with input sequence length. To reduce this computational bottleneck, researchers have developed efficient transformer architectures that seek to reduce sequence lengths via methods such as downsampling, low rank approximations, and locality sensitive hashing (Dai et al., 2020; Kitaev et al., 2020; Wang et al., 2020b; Xiong et al., 2021).

We examine the performance of the Funnel Transformer which applies average pooling to perform sequence length reduction every 4 layers. The model achieves comparable accuracy to BERT on downstream tasks while requiring 42% fewer total MAC operations through sequence length reduction. This model achieves similar downstream task performance to BERT-Base and trains 33% faster based on wall-clock time.

**Analysis** While Funnel Transformer reduces total FLOPs and obtains substantial speedups in large-scale training, this speedup does not translate to increased speed of inference as seen in Figure 7. In practice, the average pooling layers used to perform sequence length reductions add additional operations to the computation graph and increase the model's framework overhead. At low batch sizes, Funnel Transformer is framework bound at a much higher latency than BERT, and remains slower even at larger batch sizes. While some architectural innovations decrease the total number of model FLOPs, some approaches increase the size of tensor operator graphs (e.g. vis-a-vis additional layers) which can ultimately increase inference latency.

### 4.2.3 Efficient Mobile Architectures

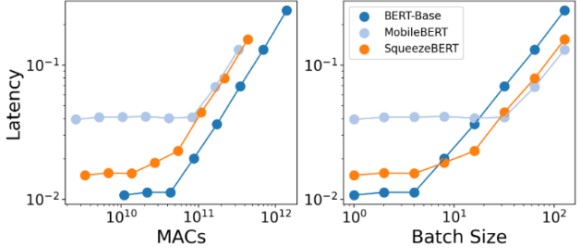

Figure 8: Latency of Transformer models using *efficient variations* of convolution and self-attention operations. All of the variants observe lower latency at large batch sizes, but have worse FLOP utilization. Efficient Transformer variants are slower than BERT at small batch sizes due to the introduction of more layers.

**Assumption** Model architectures that target deployment on edge devices, such as mobile phones, are often designed to decrease the total number of FLOPs or MACs under the assumption that such reductions translate to decreased latency. Towards this end, operations such as grouped convolutions (Zhang et al., 2018; Iandola et al., 2020), inverted residual bottlenecks (MBConvs) (Sandler et al., 2018; Sun et al., 2020), and squeeze and excitation layers (Iandola et al., 2016) have all been proposed as substitutes for dense convolution, self-attention, and linear operations. However, in practice these operations often lack the highly optimized framework and hardware support developed for more standard operations and as a result exhibit higher per-flop latency and poor memory utilization.

We examine this assumption with models that use grouped convolutions in SqueezeBERT (Iandola et al., 2020) inverted bottleneck layers and Mobile BERT (Sun et al., 2020).

**Analysis** To achieve comparable accuracy on downstream language tasks with low-FLOP operations, efficient BERT variants require much deeper model architectures which results in much higher fixed framework overhead as seen in Figure 8. Additionally, these models exhibit worse FLOP per second due to poor memory utilization compared to conventional dense linear and convolution operations. These operations can lead to slowdowns in deployment settings where depthwise and pointwise convolutions may have limited hardware and framework backend support.[3]

### 4.3 Hardware Considerations

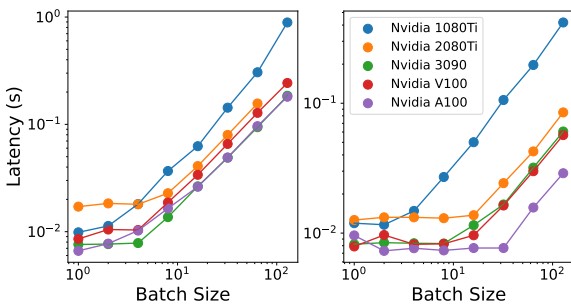

Figure 9: Framework overhead occurs across generations of GPU hardware, with increasing prominence as hardware speeds increase with newer generations.

In Figure 9, we observe that framework-bounded behaviors during inference across multiple gener-

---

[3]Pointwise convolutions are often memory-bound: `https://pytorch.org/tutorials/recipes/recipes/tuning_guide.html`

ations of consumer, workstation, and datacenter Nvidia GPUs. As the speed of the accelerator increases, the relative execution speed of the compute kernels decreases while the total framework overhead due to CPU kernel dispatch operations remains constant. We observe that GPUs that lack Tensor Core support for half precision operations, such as the 1080Ti, are notably slower and less framework-bound than newer GPUs. Conversely, this leads faster GPUs, such as the RTX 3090 and the A100, to remain framework bound at larger batch sizes for both ResNet-50 and BERT. These observations indicate that framework bounds on model execution will continue to worsen as hardware improves unless deep learning frameworks are commensurately improved. For example, BERT-Base is not framework bound for older 1080Ti GPUs but is on newer 3090 GPUs.

Framework boundedness is mainly caused by bottlenecks due to CPU dispatch operations. As a result, the fixed latency for framework-bound models is a product of the entire system configuration, dependent on hardware specifications such as CPU and memory speeds. In contrast, the latency of compute-bound models is mainly determined by the properties of kernels: operation type, kernel size, and their supported GPU FLOPs per second — with faster GPUs yielding faster execution. Additional details on hardware system configurations are provided in Appendix A.

## 5 Discussion

**Computational graph optimizations and compilation improve latency.** Removal of host language dependencies and graph optimizations provides substantial speedups over eager frameworks for inference at low batch sizes. However, feature completeness for operators and control flow varies across graph optimizers and compilers. For example, the FNet architecture (Lee-Thorp et al., 2021) relies on FFTs as a deterministic swap-in for self-attention. FFT operations are not currently supported by ONNX or TorchScript As expected, FNet executed in PyTorch outperforms BERT executed in ONNX despite less framework overhead and numerous static optimizations – with a 10.31% speedup at batch size 1. For improvements in research to translate to deployment, additional investment can be directed towards support for complex control flow and operators in inference runtimes.

**Dynamic computational graphs can be faster for input sentences with variable lengths.** Dynamic computational graphs can leverage input sparsity to reduce latency when processing variable length text. For example, PyTorch with sparse tensor optimizations reduces the latency of static CUDA graphs by 80.56% at batch size 128 when processing sparse inputs.

**At large input sizes, framework overhead from graph operations is negligible.** For batch sizes larger than 16, we find that there is minimal latency difference across models, inference runtimes, and frameworks. In the compute-bound regime, number of FLOPs is still a poor latency predictor due to variable execution time for different operations. For example, efficient mobile architectures that depend on inverted-residual layers are memory inefficient and are much slower per-FLOP than standard convolution and linear layers.

**For framework-bound models, model depth is a reasonable proxy for latency.** Number of floating point operations is a poor indicator of latency in a framework-bound setting, as total runtime is generally constant and tied to framework overheads and the size of the computational graph. In framework-bound models, the size of the computational graph is related to model depth.

**Estimations of latency for models deployed in production settings must account for their target framework and hardware platform.** Model development frequently occurs using eager execution research frameworks. However, deployment often occurs in inference runtimes and on mobile devices or specialized hardware. This misalignment can mislead the development of "efficient" models and result in claimed gains that do not translate to real-world deployment settings. As such, researchers should be clear in specifying the setting of their "efficiency" gains, such as the target frameworks and hardware platform, when developing new methods. For example, techniques such as hardware-aware neural architecture search which leverage direct latency measures must also control for framework choices to account for this mismatch.

**Throughput and input size can be increased at minimal cost for framework-bound models.** For a given model, latency is constant regardless of batch size until compute kernels saturate and exceed CPU launch costs. If computation is bot-

| SA Dim | FC Dim | Batch | Seq | Latency | TP |
|---:|---:|---:|---:|---:|---:|
| 768 | 3072 | 1 | 128 | 0.0136 | 0.0136 |
| 768 | 3072 | 4 | 128 | 0.0134 | 0.0034 |
| 768 | 3072 | 1 | 512 | 0.0134 | 0.0134 |
| 1536 | 6144 | 1 | 128 | 0.0134 | 0.0134 |

Table 1: Latency and throughput (TP) of BERT PyTorch models on RTX-8000. Scaling along batch sizes and model width shows no increase in latency.

tlenecked by framework overhead, the batch size or input size can be increased without increases in overall runtime. In practice, this can lead to the processing of larger batch sizes and sequence lengths at no additional latency cost until kernel operations saturate framework overhead.

**Model width can be increased at no cost for framework-bound models.** For a given batch size, individual layers of a framework bound model can be made wider by increasing hidden dimension or filter size without impacting latency. Wider models are known to exhibit better task performance due to increased parameter count and expressivity (Zagoruyko and Komodakis, 2016).

Model designers can leverage wider architectures with their target inference framework and hardware setting in mind to achieve higher utilization. For example, models processing few examples during inference can leverage wider layers to avoid framework bottlenecks. See Table 1.

**Using higher-performing hardware does not necessarily improve end-to-end performance.** Framework overhead limits the impact of improved hardware as it limits utilization. This trend will continue as ML-specific hardware advances without efforts to address software bottlenecks. For example, single-example inference with both BERT is slower using an A100 than using a V100 GPU despite a 2.75x increase in peak computational throughput.

## 6 Conclusion

We conduct an extensive study of neural networks from the convolutional and transformer architecture paradigms across a variety of software and hardware platforms. We show that inference performed with these large neural networks, which was previously assumed to be compute bounded, is in fact limited by overhead incurred by deep learning frameworks. While wider transformer architectures (e.g. BERT-Base) exhibit less boundedness behaviors than narrower, deeper CNNs (e.g.

ResNet-50), we show that all models exhibit framework boundedness. Additionally, we observe that these inefficiencies are becoming more apparent as hardware accelerator speeds increase.

We introduce the concept of the *framework tax* to describe when improvements in hardware speed and reductions in required computation fail to translate to speedups in model latency and throughput due to bottlenecks incurred by deep learning frameworks. We hope that these observations raise awareness of the impact and limitations created by choice of deep learning frameworks on model development and deployment.

## 7 Limitations

In this work, we study the inference efficiency of speech and language models using GPU hardware accelerators. While GPUs are the most common general purpose hardware accelerators, there exist domain specific architectures such as Google TPU's, GraphCore IPUs, and custom ASICs which present additional settings for future investigation. Additionally, our study evaluates efficiency via model latency and our claims do not necessarily translate to other metrics of efficiency, such as power consumption or power output.

As models continue to scale in size, they often require model or data parallelism techniques that require computation across several nodes which introduce overhead from multi-device synchronization and network communication. Additionally, we do not study latency in the training setting where the per-layer computation is larger due to the computation of gradients and losses.

## Acknowledgements

This work was supported in part by a grant from the National Science Foundation Graduate Research Fellowship Program under Grant No. DGE2140739. We thank the anonymous reviewers for their valuable feedback. We would also like to thank lab members and faculty for helpful feedback during discussions and revisions, including: Daniel Fried, Han Guo, Jeremiah Milbauer, Sanket Vaibhav Mehta, and Saujas Vaduguru.

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

## A  Hardware Platforms

Full details for the various hardware platforms and GPUs used for evaluation in Section 4.3 are described in Table 2 .

## B  Hardware and Utilization

In Table 3, we examine GPU hardware utilization of various frameworks in terms of the percentage of SMs active (i.e. number of compute units utilized) and warp occupancy (i.e. amount of time each compute unit is active). Graph compilations and kernel serialization reduce framework overhead and increase hardware utilization. All frameworks observe suboptimal hardware utilization

## C  Additional Experiments: Natural Language Generation

In Figure 10, we examine the GPT-2 decoder-only transformer model architecture and observe that as with encoder-only architectures generative models encounter framework boundedness for small input sizes. Evaluations are performed across batch sizes from 1 to 128 and a fixed generation length of 16 tokens.

| CPU | GPU | GPU Arch | Core Count | Tensor Cores | Clock Rate (GHz) | Memory (GB) | Mem BW (GB/s) | FP16 TFLOPS |
|---|---|---|---|---|---|---|---|---|
| **Intel Xeon Silver 4110** | **1080Ti** | Pascal | 3584 | - | 1.38 | 11 | 484.4 | 22.78 |
| **Intel Xeon Gold 6242** | **V100** | Volta | 5120 | 640 | 1.23 | 32 | 897 | 28.26 |
| **Intel Xeon E5-2630** | **2080Ti** | Turing | 3584 | 544 | 1.35 | 11 | 616 | 26.90 |
| **Intel Xeon E5-2630** | **RTX-8000** | Turing | 4608 | 576 | 1.40 | 48 | 672 | 32.62 |
| **AMD EPYC 7282** | **3090** | Ampere | 10496 | 328 | 1.35 | 24 | 936 | 35.89 |
| **Intel Xeon Silver 4110** | **A6000** | Ampere | 10752 | 336 | 1.41 | 48 | 768 | 38.71 |
| **Intel Xeon 8339HC** | **A100** | Ampere | 6912 | 432 | 1.215 | 40 | 1935 | 77.97 |

Table 2: Details on hardware platforms used in our experiments, ordered by Nvidia microarchitecture generation.

| Framework | Graph Compilation | Kernel Serialization | Latency | SMs Active | Warp Occupancy |
|---|---|---|---|---|---|
| PyTorch | None | None | 10.54 ms | 2.6% | 0.9% |
| PyTorch with TorchScript | Just-in-Time | None | 6.14 ms | 18.5% | 3.0% |
| PyTorch with CUDA Graphs | None | Yes | 2.82 ms | 57% | 9.2% |
| ONNX RT | Ahead-of-Time | None | 2.56 ms | 22.3% | 9.5% |
| ONNX RT with CUDA Graphs | Ahead-of-Time | Yes | 2.11 ms | 59% | 20.3% |

Table 3: Comparison of SM Activity Across Frameworks for BERT-Base at batch size 1.

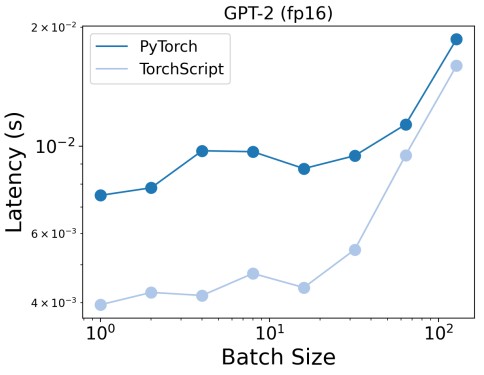

Figure 10: Latency of generative language models for varying batch sizes.

# D  Additional Experiments: Vision

In Figures 11 and 12, we examine vision models utilizing "efficient" model design choices through scaling and efficient operation variants. Image inputs are simulated by randomly generating three-channel 224 x 224 RGB images. As with language models, deeper models introduce additional framework overhead and low-FLOP alternatives are more framework bound.

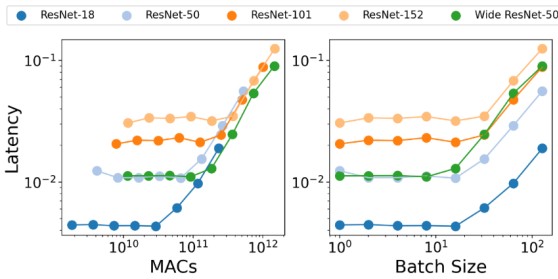

Figure 11: Latency of vision models that scale model depth and number of hidden dimensions.

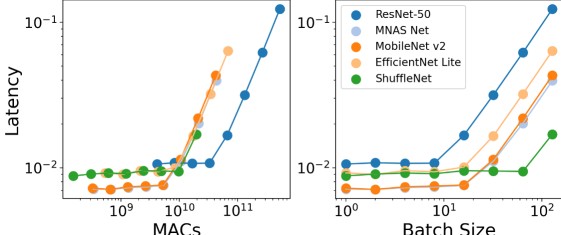

Figure 12: Latency of vision models using *efficient variations* of convolution and self-attention operations.

# E  Additional Experiments: Speech

In Figure 13, we examine the behavior of the WavLM (Chen et al., 2022) model which consists of a CNN encoder followed by transformer encoder layers. Audio inputs are simulated as 2 second sequences sampled at 16 kHz to create 32,000-dimensional floating point inputs. In Figure 13, we observe that WavLM exhibits framework bound behavior but quickly transitions to being compute-bound due to the large audio sequence lengths.

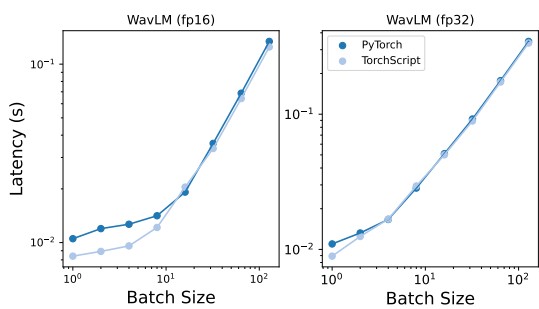

Figure 13: Transformer-based speech models exhibit framework boundedness but transition to compute-bound at small batch sizes due to long sequence lengths.