# OpenReview forum: "The Framework Tax: Disparities Between Inference Efficiency in NLP Research and Deployment"
_EMNLP/2023/Conference — EMNLP 2023 Main_

### Official Review · Reviewer_GHwB · 2023-08-01

**Soundness:** 4

**Excitement:**

4: Strong: This paper deepens the understanding of some phenomenon or lowers the barriers to an existing research direction.

**Paper Topic And Main Contributions:**

The authors introduce a thorough evaluation on estimating the cost of deep learning model frameworks vs the actual computation which occurs on CPU and GPU. as a part of their study, the demonstrate that many improvements in FLOPs lead to no improvements in wall clock, which they attribute to the frameworks used. They then go on to posit that the framework tax will increase over time despite improvements in hardware speed.
Moreover, this paper introduces well-formulated tidbits that can be useful to researchers deploying language models such as:
1. At small batches inference is compute bound but as batch size scales models become framework bound.
2. The use of specialized models are often not faster in wall clock as they are memory bound.
3. Older GPUs are not framework bound but novel more expensive GPUs are.

**Questions For The Authors:**

1. Why did you not look at quantization?
2. Do you see any difference across fp16 to bf16?

**Reasons To Accept:**

1. Paper provides an easy-to-read, thorough evaluation of the impacts to model latency and how frameworks and computations play together.
2. Paper evaluates many sub-components of latency using different models, batch sizes, etc.
3. Evaluation is on many types of computing hardware

**Reasons To Reject:**

1. The paper does not analyze any generative models such as Llama.
2. The paper does not analyze quantized models in either int8 or int4.
3. The evaluation does not explore optimized runtimes like Faster Transformers, which commonly lead to 10x speedup, especially in small models.
4. The paper does not study the impact of sequence length on framework vs compute bounding

**Reproducibility:**

4: Could mostly reproduce the results, but there may be some variation because of sample variance or minor variations in their interpretation of the protocol or method.

**Reviewer Confidence:**

4: Quite sure. I tried to check the important points carefully. It's unlikely, though conceivable, that I missed something that should affect my ratings.

---

> ### Author Rebuttal · Authors · 2023-08-29
>
> We appreciate the reviewer’ positive feedback and agree that our insights on bottlenecks introduced by deep learning frameworks can provide immediately useful guidance in the deployment of efficient NLP systems! We hope that our work can help researchers better understand practical bottlenecks and ensure that efficiency gains attained in research settings translate to deployed NLP systems.
>
> ## Questions and Concerns
>
> `The paper does not analyze any generative models such as Llama.`
>
> We focus our analysis on BERT-like sequence classifications models to investigate the effectiveness of efficient model architecture designs which  have largely targeted encoder-only models (e.g. MobileBERT, SqueezeBERT, DistilBERT, etc).
>
> In preliminary experiments with decoder-only LMs, we find that the framework tax also leads to inference bottlenecks of generative models.  For sequence generation, we observe framework bound behaviors with roughly fixed latency for batch sizes smaller than 16.
>
> GPT-2 Base in PyTorch (SeqLen = 16 Tokens):
> | Batch Size | Latency (ms) |
> |-|-|
> |1|	6.993
> |2|	7.987
> |4|	8.629
> |8|	8.241
> |16|	8.791
> |32|	9.237
> |64|	13.72
> |128| 18.78
>
> `The paper does not analyze quantized models in either int8 or int4. Why did you not look at quantization?`
>
> We primarily focus on optimizations from computational graph compilation and sparsity, as quantization is not currently natively supported for GPU operations in PyTorch nor by ONNX Runtime’s CUDA Execution Provider.
>
> Ref: https://pytorch.org/docs/stable/quantization.html#frequently-asked-questions
>
> `The evaluation does not explore optimized runtimes like Faster Transformers, which commonly lead to 10x speedup, especially in small models.`
>
> In Section 4.1, we investigate sparse transformer and nested tensor accelerations with PyTorch’s BetterTransformer optimizations. BetterTransformer provides FlashAttention and FastPath optimizations which are similar to the optimizations provided by Nvidia’s FasterTransformer.
>
> As with both the JiT and AoT runtimes, we observe that FasterTransformer yields reduces latency but still suffers from framework boundedness at lower batch sizes. We will be sure to clarify implementation details in future revisions.
>
> `Do you see any difference across fp16 to bf16?`
>
> With an Nvidia RTX-8000 GPU, we observe that for both float16 and bfloat16, inference latency exhibits framework bounded behavior.  However, inference with bfloat transitions to compute boundedness at smaller batch sizes. This discrepancy can likely be attributed to the lack of bfloat16 support by the Turing generation of Nvidia GPUs which results in slower computation.
>
> Ref: https://www.nvidia.com/en-us/data-center/tensor-cores/
>
> | Batch Size | Float16 Latency (ms) | BFloat Latency (ms) |
> | - | - | -|
> 1 | 6.24| 12.75
> 2 | 8.58 | 11.75
> 4 | 8.24  | 15.18
> 8 | 8.26  | 24.95
> 16 | 9.28 | 49.17
> 32 | 18.03 | 92.93
> 64 | 32.5 | 184.65
> 128 | 63.8 |  365.03
>
> `The paper does not study the impact of sequence length on framework vs compute bounding`
>
> While we primarily study the impact of batch size scaling, framework boundedness is not only limited to small batch sizes but also bottlenecks inference on short sequences. In additional experiments (below), we observed the latency of BERT-Base for various sequence lengths at batch size 128 and observe boundedness for sequences shorter than 8 tokens.
>
> |Seq Len |	Latency for BS=128  (ms)
> |--|--|
> 1	|	6.05
> 2|	        8.39
> 4|	         8.40
> 8|	8.30
> 16|	9.73
> 32|	17.07
> 64|	30.58
> 128 | 63.63

---

### Official Review · Reviewer_fRoK · 2023-08-04

**Soundness:** 4

**Excitement:**

4: Strong: This paper deepens the understanding of some phenomenon or lowers the barriers to an existing research direction.

**Paper Topic And Main Contributions:**

Despite improvements in NLP model architectures and hardware accelerators, there's a discrepancy between the computational efficiency improvements and the real-world (wall-clock) inference latency. The paper attributes this discrepancy mainly to overheads introduced by deep learning frameworks, terming it the "framework tax".

Key contributions:

1. Demonstrated that advancements in neural networks and hardware don't necessarily reduce inference latency due to the "framework tax, " which increases as hardware speeds up.

2. Investigated the effects of model design decisions, deep learning framework paradigms (eager execution, just-in-time, and ahead-of-time compilation), and hardware platforms on total model latency.

3. Evaluated transformer and convolutional neural network model performance across various environments (PyTorch, TorchScript, ONNX) and GPUs. Findings revealed that framework overhead dominates at small batch sizes, leading to proxies like MACs and parameter counts not predicting inference latency effectively.

4. Suggested using ahead-of-time inference runtimes for small batch sizes to reduce overhead. For those using eager execution, increasing model width or batch size doesn't cost extra latency, providing extra flexibility in model design.

5. Highlighted the divergence between research settings and deployment, emphasizing that researchers should specify efficiency gains contextually (frameworks, hardware).

6. Pointed out that more performant hardware doesn't necessarily equate to improved end-to-end performance because of the "framework tax".

**Questions For The Authors:**

A. The results focus heavily on PyTorch, TorchScript, and ONNX. While these are undoubtedly popular, what about other major frameworks, such as TensorFlow 2.0? Have you observed any similar disparities in those frameworks?

B. In the "Experiments" section, you've focused on specific GPU microarchitectures (Pascal, Turing, Ampere). While this selection covers a broad range, it would be enlightening to understand the rationale behind excluding certain GPU architectures or why these specific architectures were chosen.

C. While the discussion emphasizes the disadvantages of the framework tax, are there potential advantages or reasons why these overheads exist in the first place? For instance, might any trade-offs related to functionality, flexibility, or developer ease-of-use justify some amount of this tax?

D. The "Limitations" section briefly touches upon other hardware accelerators like TPUs and ASICs but doesn't delve into the potential implications of the framework tax in those settings. Given the growing popularity of domain-specific architectures for AI, could you speculate or provide preliminary observations about how the framework tax might manifest in those contexts?

**Reasons To Accept:**

1. The paper addresses the significant yet overlooked problem of the "framework tax," where improvements in hardware and model architectures do not necessarily translate to reduced inference latencies.
2. Through systematic investigation, the authors evaluate different model architectures across multiple frameworks and GPUs, providing a robust and varied assessment of the phenomenon.
3. The paper offers actionable insights and recommendations, like the suggestion to use static or ahead-of-time inference runtimes for small batch sizes, that can greatly benefit NLP researchers and practitioners.
4. By highlighting the disparities between NLP research and real-world deployment, the paper fosters an understanding that could bring research and practice closer, ensuring that efficiency gains in the research stage translate well in deployment.

Benefits to the NLP community:
1. Researchers and developers will become more aware of the overhead introduced by deep learning frameworks, potentially influencing their choice of frameworks and methodologies.
2. Model designers can use the findings to create both computationally efficient models and have reduced inference latencies, leading to faster real-world applications.
3. For developers with existing deployments, the insights from this paper can guide adjustments to current systems for better performance.
4. The paper sets the stage for future research on framework optimization, model-hardware co-design, and a deeper understanding of the intricate play between model design, hardware, and frameworks.

**Reasons To Reject:**

1. The paper focuses predominantly on GPUs for evaluation. Given the wide variety of hardware accelerators used in the industry, such as TPUs, IPUs, and custom ASICs, the study's findings might not generalize across all hardware platforms.

2. The paper measures efficiency primarily through model latency. This ignores other crucial efficiency aspects such as power consumption, cost-effectiveness, and overall throughput.

3. While the "framework tax" is an interesting observation, the paper treats all NLP models uniformly. Given the diversity of NLP tasks and associated models, it's possible that not all models will exhibit similar "framework tax" phenomena.

4. While the paper mentions that certain operations and design decisions could potentially reduce latency or framework overhead, some of these claims lack empirical results to back them up, relying on theoretical reasoning alone.

Risks of presenting at the conference:

1. Presenting the idea of a "framework tax" might oversimplify the challenges faced in deploying NLP models. Attendees might walk away thinking that framework overhead is the only significant issue, which might not be the case.

2. Since the paper focuses heavily on GPUs, it might unintentionally endorse or bias attendees towards GPU-based architectures, neglecting other emerging and efficient architectures.

3. If attendees focus too much on the "framework tax" and neglect other aspects of model efficiency, it might misdirect research and development efforts towards only one aspect of the problem.

4. Given the emphasis on the inefficiencies of existing frameworks, the paper might receive pushback or defensive reactions from developers or proponents of the said frameworks, leading to polarized discussions.

**Reproducibility:**

4: Could mostly reproduce the results, but there may be some variation because of sample variance or minor variations in their interpretation of the protocol or method.

**Reviewer Confidence:**

4: Quite sure. I tried to check the important points carefully. It's unlikely, though conceivable, that I missed something that should affect my ratings.

---

> ### Author Rebuttal · Authors · 2023-08-28
>
> Thank you to the reviewer for their positive comments and comprehensive feedback. As the reviewer notes, there are many components and modeling choices that contribute to the design of efficient NLP systems. While the framework tax and its associated framework overhead are by no means the sole bottleneck on efficiency, we believe that our study provides valuable analysis of common workloads and identifies an often ignored bottleneck not captured by existing metrics (e.g. FLOPs/MACs, parameter counts).
>
> ## Questions & Concerns
> `While the "framework tax" is an interesting observation, the paper treats all NLP models uniformly. Given the diversity of NLP tasks and associated models, it's possible that not all models will exhibit similar "framework tax" phenomena.`
>
> Although there exist settings in which other factors  may limit system latency, such as user interaction in human in the loop systems, many modern NLP systems rely on large neural network transformer-based models – a setting where we show framework overhead can often impose bottlenecks on inference efficiency. Additionally, we show that framework boundedness is not limited to encoder-only models and with CNNs and hybrid CNN-transformer architectures exhibiting the same behaviors in Appendix C and D.
>
> Furthermore, in additional experiments with decoder-only GPT-2 (for maxlen 16), we find that framework overhead also bottlenecks seq2seq language generation small batch sizes
>
> | Batch Size | Latency (ms) |
> |-|-|
> |1|	6.993
> |2|	7.987
> |4|	8.629
> |8|	8.241
> |16|	8.791
> |32|	9.237
> |64|	13.72
> |128| 18.78
>
> `The paper measures efficiency primarily through model latency. This ignores other crucial efficiency aspects such as power consumption, cost-effectiveness, and overall throughput.`
>
> While other efficiency measures are important in assessing the efficiency of NLP systems, inference latency is a common real-world bottleneck in deployment and frequently reported measurement among existing efficiency benchmarks [3, 4]. We leave these investigations as potential directions for future work as efficiency cost indicators are often uncorrelated in practice [2].
>
> `The results focus heavily on PyTorch, TorchScript, and ONNX. While these are undoubtedly popular, what about other major frameworks, such as TensorFlow 2.0? Have you observed any similar disparities in those frameworks?`
>
> We select PyTorch (eager), TorschScript (JiT), and ONNX RT (AoT) as representative examples of frameworks from various design paradigms. As an exhaustive evaluation of all deep learning frameworks is beyond the scope of this work, our goal is to provide insight into the overhead associated with each paradigm. As TF2.0 is based around the eager execution, we expect to observe high framework overhead as observed with eager execution PyTorch.
>
> ` In the "Experiments" section, you've focused on specific GPU microarchitectures (Pascal, Turing, Ampere). While this selection covers a broad range, it would be enlightening to understand the rationale behind excluding certain GPU architectures or why these specific architectures were chosen.`
>
> The Pascal, Turing, and Ampere microarchitectures and the associated GPUs are representative of publicly available consumer  (eg. 1080 Tis, 2080Ti, 3090) and enterprise  (e.g. v100, RTX8000, A6000)  GPUs at the time experiments were conducted. They were selected to reflect increases in hardware speed (e.g. via increases in number of CUDA core counts, clock speed, etc) as well as the introduction of new computing technology (e.g. the introduction of tensor cores in the Turing generation).
>
> `While the discussion emphasizes the disadvantages of the framework tax, are there potential advantages or reasons why these overheads exist in the first place? For instance, might any trade-offs related to functionality, flexibility, or developer ease-of-use justify some amount of this tax?`
>
> As the reviewer notes, efficiency tradeoffs are often made to increase usability. For example, PyTorch allows for rapid prototyping via eager construction of computational graphs in Python – but incurs host language overheads and forgoes optimizations from graph compilation. While these tradeoffs are often beneficial to research velocity, one of our goals in highlighting these sources of overhead to motivate the design of frameworks that are both lightweight and usable.
>
> `Since the paper focuses heavily on GPUs, it might unintentionally endorse or bias attendees towards GPU-based architectures, neglecting other emerging and efficient architectures.`
>
> `The "Limitations" section briefly touches upon other hardware accelerators like TPUs and ASICs but doesn't delve into the potential implications of the framework tax in those settings. Given the growing popularity of domain-specific architectures for AI, could you speculate or provide preliminary observations about how the framework tax might manifest in those contexts?`
>
> In our study we primarily focus on the GPUs due to their widespread availability and high usage as deep learning hardware accelerators. Unfortunately, we did not have access to domain specific hardware. Based on prior performance analyses of custom hardware accelerators [1], we expect that CPU and framework overhead would also impose bottlenecks on execution with TPUs as they follow a similar design based around a host CPU and a TPU compute device.
> Additionally, in additional experiments on CPU-only devices where there is no need for operation dispatch to GPUs, we find that framework overhead is less pronounced.
>
> ### References
>
> [1] Wang, Yu, Gu-Yeon Wei, and David Brooks. "A systematic methodology for analysis of deep learning hardware and software platforms." Proceedings of Machine Learning and Systems 2 (2020): 30-43.
>
> [2] Dehghani, Mostafa, et al. "The Efficiency Misnomer." International Conference on Learning Representations. 2021.
>
> [3] Peng, Hao, et al. "Efficiency Pentathlon: A Standardized Arena for Efficiency Evaluation." arXiv preprint arXiv:2307.09701 (2023).
>
> [4] Reddi, Vijay Janapa, et al. "Mlperf inference benchmark." 2020 ACM/IEEE 47th Annual International Symposium on Computer Architecture (ISCA). IEEE, 2020.

---

### Official Review · Reviewer_aBkf · 2023-08-06

**Soundness:** 4

**Excitement:**

4: Strong: This paper deepens the understanding of some phenomenon or lowers the barriers to an existing research direction.

**Paper Topic And Main Contributions:**

This paper presents a series of case studies to analyze the framework tax phenomenon existing in deep learning frameworks and its effect on model efficiency. Overall, there is no comparison with other studies.

**Questions For The Authors:**

The contribution should be highlighted at the end of Introduction.

**Reasons To Accept:**

Presents a series of case studies to analyze the framework tax phenomenon.

**Reasons To Reject:**

1. There is no comparison with other studies.


**Reproducibility:**

3: Could reproduce the results with some difficulty. The settings of parameters are underspecified or subjectively determined; the training/evaluation data are not widely available.

**Reviewer Confidence:**

3: Pretty sure, but there's a chance I missed something. Although I have a good feel for this area in general, I did not carefully check the paper's details, e.g., the math, experimental design, or novelty.

---

> ### Author Rebuttal · Authors · 2023-08-28
>
> We appreciate the reviewer’s positive assessment of our work. As the reviewer notes, this work is the first to our knowledge to systematically examine the impact of deep learning frameworks on the efficiency of NLP systems. As we observe in our Related Work, there exist performance analyses of deep learning-based systems in other fields such as recommendation systems and simulation-based reinforcement learning [1, 2]. For the final revision, we will be sure to more clearly include comparisons with these works and to emphasize our main contribution in the introduction.
>
> ### Reference
>
> [1] James Gleeson, Moshe Gabel, Gennady Pekhimenko, Eyal de Lara, Srivatsan Krishnan, and Vijay Janapa Reddi. 2021. Rl-scope: Cross-stack profiling for deep reinforcement learning workloads. Proceedings of Machine Learning and Systems, 3:783–799.
>
> [2] Zhongyi Lin, Louis Feng, Ehsan K Ardestani, Jaewon Lee, John Lundell, Changkyu Kim, Arun Kejariwal, and John D Owens. 2022. Building a performance model for deep learning recommendation model training on gpus. arXiv preprint arXiv:2201.07821.

---

### Meta-Review · Area_Chair_mfoF · 2023-09-08

**Recommendation:** 5

**Metareview:**

This paper studies why improvements in throughput and FLOPs of recent efficient models and hardware architectures do not directly translate to reductions in model inference latency. The paper attributes this gap to the overhead imposed by some deep learning frameworks, to which it refers as "framework tax." Overall, the reviewers unanimously rate this paper as strong both with regards to soundness and excitement.

The reviewers praise this paper for the broad insights it provides on a practically relevant topic. They also praise the novelty and systematic investigation, and note that this can be widely useful for the NLP community.

Most of the reviewer criticism centers around extending the study to other accelerators, models (beyond encoder-only transformers), and other aspects such as to investigate different sequence lengths. While the present paper does not cover all possible experiment settings, it nevertheless has been praised for its broadness by the reviewers. This suggests that the identified gaps could motivate further work on this subject in the future. Moreover, the authors provide preliminary insights in their responses on some of the settings requested by the reviewers, such as decoder-only transformers, and regarding sequence length. These results could provide additional value to the submission.

---

### Decision · Program_Chairs · 2023-10-07

**Decision:**

Accept-Main

**Comment:**

This paper studies why improvements in throughput and FLOPs of recent efficient models and hardware architectures do not directly translate to reductions in model inference latency. The paper attributes this gap to the overhead imposed by some deep learning frameworks, to which it refers as "framework tax." Overall, the reviewers unanimously rate this paper as strong both with regards to soundness and excitement.

The reviewers praise this paper for the broad insights it provides on a practically relevant topic. They also praise the novelty and systematic investigation, and note that this can be widely useful for the NLP community.

Most of the reviewer criticism centers around extending the study to other accelerators, models (beyond encoder-only transformers), and other aspects such as to investigate different sequence lengths. While the present paper does not cover all possible experiment settings, it nevertheless has been praised for its broadness by the reviewers. This suggests that the identified gaps could motivate further work on this subject in the future. Moreover, the authors provide preliminary insights in their responses on some of the settings requested by the reviewers, such as decoder-only transformers, and regarding sequence length. These results could provide additional value to the submission.